# DNA Glycosylases Define the Outcome of Endogenous Base Modifications

**DOI:** 10.3390/ijms241210307

**Published:** 2023-06-18

**Authors:** Lisa Lirussi, Hilde Loge Nilsen

**Affiliations:** 1Department of Clinical Molecular Biology, Institute of Clinical Medicine, University of Oslo, 0318 Oslo, Norway; 2Section of Clinical Molecular Biology (EpiGen), Akershus University Hospital, 1478 Lørenskog, Norway; 3Department of Microbiology, Oslo University Hospital, 0424 Oslo, Norway; 4Unit for Precision Medicine, Akershus University Hospital, 1478 Lørenskog, Norway

**Keywords:** endogenous DNA damage, single base modifications, epigenetic marks, 5-methylcytosine (5-mC), 5-hydroxymethylcytosine (5-hmC), 5-hydroxymethyluracil (5-hmU), Base Excision Repair (BER), DNA glycosylases, SMUG1

## Abstract

Chemically modified nucleic acid bases are sources of genomic instability and mutations but may also regulate gene expression as epigenetic or epitranscriptomic modifications. Depending on the cellular context, they can have vastly diverse impacts on cells, from mutagenesis or cytotoxicity to changing cell fate by regulating chromatin organisation and gene expression. Identical chemical modifications exerting different functions pose a challenge for the cell’s DNA repair machinery, as it needs to accurately distinguish between epigenetic marks and DNA damage to ensure proper repair and maintenance of (epi)genomic integrity. The specificity and selectivity of the recognition of these modified bases relies on DNA glycosylases, which acts as DNA damage, or more correctly, as modified bases sensors for the base excision repair (BER) pathway. Here, we will illustrate this duality by summarizing the role of uracil-DNA glycosylases, with particular attention to SMUG1, in the regulation of the epigenetic landscape as active regulators of gene expression and chromatin remodelling. We will also describe how epigenetic marks, with a special focus on 5-hydroxymethyluracil, can affect the damage susceptibility of nucleic acids and conversely how DNA damage can induce changes in the epigenetic landscape by altering the pattern of DNA methylation and chromatin structure.

## 1. Endogenous DNA Damage

Tomas Lindahl founded a new research topic when he experimentally measured the rate of spontaneous decay of DNA at physiological conditions [1]. The rates he determined for cytosine deamination are part of the standard literature [1], even tough technological development has provided the opportunity for highly sensitive quantitative measurements of base modifications.

Methylation and other single base modifications widely occur in nucleic acids. The same modification can lead to cytotoxic and/or carcinogenic DNA damages when deriving from endogenous and exogenous insults or can impact gene transcription and fine-tune gene expression regulation when tightly programmed, depending on the cellular context, nucleotide composition, and spatial organization. Some examples of these are the oxidation of guanine, cytosine deamination to uracil, and the oxidation intermediates of 5-hydroxymethylcytosine (5-hmC) and 5-hydroxymethyluracil (5-hmU). The discrimination between when a modification is an instance of DNA damage or an epigenetic mark then represents a challenge, as the cell needs to exert different programs in response to the same chemical modification. While DNA damage response and repair processes ensure genome integrity, correcting aberrations in the DNA structure, the epigenetic program recruits additional factors for the modulation of gene expression (Figure 1). 

Incorrectly interpreted signals may lead to deleterious consequences for the cells. Thus, a precise regulation and coordinate response is required. The simultaneous presence of chemically and enzymatically modified bases suggests an intersection between DNA damage, DNA repair (especially BER proteins), and epigenetic regulation. DNA glycosylases are key regulators of gene expression due to their capability to recognize small chemical modifications acting as DNA sensors: upon detection of these modified bases, they actively remove them via DNA repair processes to maintain the (epi)genome integrity. Interestingly, the converse is also true. Epigenetic marks such as 5-mC influence DNA repair and genome stability.

## 2. Endogenous DNA Damage and the Base Excision Repair (BER) Pathway

Tomas Lindahl isolated the proteins that excise uracil, the product of cytosine deamination [2], and his lab was instrumental in reconstituting the entire repair pathway on naked DNA [3] and nucleosomes [4]. The integrity of the cellular DNA is continuously challenged by endogenous DNA damage, generated by the normal cellular metabolism. This includes DNA oxidation, alkylation, and deamination and the formation of abasic (AP) sites. Depending on the genomic location where these modifications occur, these single base damages can also affect the fidelity and the progression of DNA and RNA polymerases, possibly leading to inaccurate transcription and interrupted elongation [5]. To preserve genome integrity, cells have evolved a complex system of DNA repair pathways to counteract the different types of DNA insults. The base excision repair (BER) pathway is able to recognize and replace single modified bases not distorting the DNA helix in a highly regulated series of steps. The first step in the BER is executed by one out of eleven known DNA glycosylases that have the capability to distinguish subtle chemical modifications on DNA bases and excise them by cleaving the *N*-glycosidic bond between the aberrant base and the deoxyribose sugar, generating an AP site [6,7,8]. The newly generated AP site is then recognized by the Apurinic/apyrimidinic Endonuclease 1 (APE1), which creates 3′-OH and 5′-deoxyribose phosphate (dRP) termini through the incision of the phosphodiester backbone. DNA polymerases fill in the gap generated, and DNA ligases (DNA Ligase I or DNA Ligase IIIa/XRCC1) create a covalent phosphodiester bond by sealing the resulting nick [6,7,8].

As many base modifications change base pairing properties, and therefore have mutagenic capacity [9], it was somewhat surprising that knockout mice deficient in DNA glycosylases, generally, are not prone to develop spontaneous cancer [10,11,12]. Unexpectedly, these knock-out mice present mild to moderately elevated mutation frequencies under normal physiological conditions. In some models, like the UNG knock-out mice, there was some spontaneous tumour formation but at low penetrance and late onset [11,12]. The absence of a tumour-prone phenotype supports the hypothesis that DNA glycosylases have evolved specialized functions beyond DNA repair associated with gene regulation, replication, and chromatin remodelling [10,11,12]. The only cancer syndromes associated with mutations in BER are MUTYH1-associated polyposis (MAP) [13,14,15] or NTHL1-associated polyposis (NAP) [16], of which the latter gives tumour-predisposition in other tissues [17]. A lot of effort has been dedicated to systematically defining the mutations arising from exogenous damage. Recently, a CRISPR-Cas9 screen explored and evaluated the consequences of endogenous DNA damage on mutational footprints and mutation signatures, defining critical genes to maintaining genome stability [18]. Interestingly, of the three DNA glycosylases analysed (OGG1, TDG and UNG), only OGG1 and UNG knock-outs were able to leave a substitution signature [18]. Whole-genome sequencing of cancer genomes has revealed mutation signatures associated with defects in BER, due to compromised DNA glycosylase activity and by shaping mutation signatures connected with AID/APOBEC family enzymes [19,20]. Oxidative damage (8-oxoG formation) produces a G > T/C > A pattern, similar to the one identified in adrenocortical cancers and neuroblastomas (SBS18) and qualitatively analogous to the mutational signature of inactivating of germline mutations in MUTYH1 (SBS36) [18,21]. Deamination of 5-mC to thymine giving C > T in CpG contexts is found in many cancers with biallelic loss of MBD4 (Methyl-CpG-binding domain 4), and it is known as SBS1 [18,22]. This signature is similar to SBS30 and is a footprint analogue to the one for NTHL1, except for the trinucleotide preference observed for NTHL1 [17,18]. 

When considering mutation signatures (in SBS2, SBS13) in tumours expressing high levels of the cytidine deaminases APOBEC3A/APOBEC3B (apolipoprotein B mRNA editing enzyme, catalytic polypeptide-like 3B and 3A) [23,24], it would be expected that UNG2 would influence the mutation spectrum. Disease-causing variants of UNG2, which give rise to Hyper IgM syndrome type V [25], appear to be underrepresented in cancer [26]. This might be explained by the observation that loss of UNG appears to affect fitness of cells expressing AID (activation-induced cytidine deaminase) [26] and even induce synthetic lethality with APOBEC3B overexpression [27]. 

## 3. Epigenetic Modifications and Their Functions

### 3.1. Epigenetic DNA Modifications

The complexity and the diversity of epigenetic regulation is achieved through post-translational modifications of DNA bases and histones, which modulate local chromatin structural changes and ultimately alter the transcription level. Its dynamicity plays an essential role in responding to different environmental conditions, and it is accomplished via a complex interaction of factors such as DNA methylation/demethylation enzymes, non-coding RNAs, and chromatin remodellers [28,29]. The modification of nucleotide bases, such as DNA methylation or oxidation, is essential for the epigenetic control of gene expression, adding an extra layer of regulation [30,31,32].

#### 3.1.1. Methylation of Cytosine and Its Products

The most abundant modification is the methylation of the cytosine residue (5-methylcytosine, 5-mC) by DNA methyltransferases (DNMTs), which affects the accessibility of genomic regions to regulatory complexes. It occurs predominantly in symmetric (CG and CHG) contexts. The outcome on gene transcription depends on the genomic location and cellular context of 5-mC. When present in promoters or enhancer regions, 5-mC has a negative effect on gene transcription, whilst gene body methylation might repress or enhance the transcriptional activity [33,34,35]. Moreover, 5-mC has been described as having a role in differentiation, genomic imprinting, X-chromosome inactivation, and suppression of transposons [5,29,36]. In addition to 5-mC, other important epigenetic marks are 5-hydroxymethylcytosine (5-hmC), 5-formylcytosine (5-fC), and 5-carboxylcytosine (5-caC), all deriving from sequential oxidation steps from 5-mC by the Ten-Eleven Translocation family (TET1-3) proteins and N^6^-methyladenine (6-mA) [37]. Writers, erasers, and readers, proteins responsible for maintaining, removing, and interpreting the epigenetic marks, are identified [38,39]. 

5-hmC controls transcription, chromatin architecture, and alternative splicing, depending on the genomic region. It can be stably found in promoters, enhancers, coding regions of actively transcribed genes, 3′-UTR, and intragenic regions [28,40,41,42,43]. 5-fC is also found in the genome and plays a role in nucleosome organization and transcription [44,45,46]. Although less expressed, the other intermediate products (5-fC and 5-caC), together with 5-hydroxymethyluracil (5-hmU), derived from 5-hmC deamination or direct oxidation of thymine by TET [47,48], are also considered epigenetic marks [5,30] for transcriptional regulation, chromatin remodelling, and recruitment of DNA repair-associated complexes due to increased flexibility and hydrophobicity of the DNA helix introduced by the modified base [5,30,49,50].

Methylation levels are dynamically controlled through active and passive DNA demethylation processes. In addition to passive demethylation, where these modified bases are diluted through replication, three mechanisms have been proposed for active demethylation. Two of them rely on TDG (thymine DNA glycosylase) and SMUG1 (single-stranded selective monofunctional uracil DNA glycosylase) for the recognition of the oxidized cytosines and on BER for the restoration of the unmodified base. TDG recognizes and removes both 5-fC and 5-caC. This excision generates an AP site that is repaired by BER restoring an unmodified cytosine. The second mechanism suggests a direct deamination of 5-hmC to 5-hmU by AID and APOBEC family proteins. The result base, 5-hmU, is then recognized by TDG or SMUG1 and finally repaired by BER [51]. Although deamination from 5-hmC to 5-hmU does occur in cells under special conditions [47], this mechanism is still debated, due to AID’s preference for single-stranded DNA and missing evidence that APOBEC enzymes deaminate 5-hmC in vivo [52,53]. Thus, 5-hmC-dependent deamination under specific conditions cannot be excluded, but it is likely not the main mechanism for the formation of 5-hmU. The last mechanism proposes a direct conversion from 5-caC to cytosine by a putative, yet to be identified decarboxylase [29,30,47,48,54,55,56,57,58] (Figure 2). 

During somatic reprogramming, 5-hmC initiates the demethylation of reprogramming enhancers and promoters, leading to the formation of induced pluripotent stem cells (iPSCs) [59]. 5-hmC deposition alone is not sufficient to initiate the iPSCs program per se, but the other oxidation products (5-fC and 5-caC), the recognition of these modifications by TDG, and the subsequent initiation of the BER are necessary for the reprogramming, suggesting a distinct function for the DNA demethylation pathway as a regulator of epigenetic identity and cell fate [59].

Although methylation does not alter the Watson–Crick base pairing [60], cytosine methylation affects DNA secondary structure in C-rich sequences through changes in DNA hydrophobicity, steric hindrance, and in the DNA’s mechanical properties [29]. Interestingly, while 5-mC stabilizes the DNA helix and reduces replication and transcription rates, the oxidation of 5-mC is able to revert these effects to the level of unmodified cytosine [29,31], suggesting an additional way by which these chemical modifications influence genomic processes via impeding or facilitating the unwinding of the DNA helix and helping in the recognition of the modified base [29]. 

#### 3.1.2. Oxidation of Guanine (8-oxoG)

The guanine base is the most reactive to oxidation, forming the well-known DNA base damage 8-oxoguanine (8-oxoG). 8-oxoG pairs with adenine and it induces G > T mutations if not promptly repaired by OGG1 (8-oxoguanine DNA glycosylase) and the BER pathway [61]. Although this deleterious effect on genomic stability, 8-oxoG has been recently added as epigenetic modification that affects transcriptional regulatory elements (G-quadruplex) at promoter regions, histone modifications, and methylation at CpG islands, ultimately regulating gene expression [61]. 8-oxoG affects the local methylation in two manners when it is located near CpG islands, and its effect depends on the cellular status and promoter sequence context. It negatively interferes with the deposition of 5-mC via the inhibition of DNMT binding. The OGG1-8-oxoG association recruits TET1 to the site, which in turn oxidizes adjacent 5-mC to 5-hmC, generating nuclear ROS (reactive oxygen species). These ROS further oxidized the DNA, forming 8-oxoG and inducing gene transcription via oxidative DNA damage response. In contrast, increased levels of 8-oxoG induced by oxidative DNA damage favours CpG island hypermethylation and chromatin condensation with the consequent silencing of damaged DNA regions [61,62].

### 3.2. Epigenetic RNA Modifications

Similar to cellular DNA, RNA is also extensively chemically modified, and such modifications occur on all types of RNAs and influence all aspects of RNA metabolism, increasing the complexity of RNA regulation [63,64,65,66,67,68]. RNA is highly susceptible and reactive to oxidation, and numerous oxidized RNA modifications are formed such as 8-oxoguanine (o^8^G), 8-oxoadenine (o^8^A), 5-hydroxyuridine, and 5-hydroxycytidine [61,69].

#### 3.2.1. RNA Methylation

Methylation is widely present in RNA and controls the structure and function of mature RNA. Most commonly, methylation occurs on the terminal guanosine of the cap (N^7^-methylguanosine in mRNA), on internal adenosine (i.e., N^6^-methyladenosine in mRNA and N^1^-methyladenosine in tRNA), and on cytosine residues (i.e., N^3^-methylcytosine in rRNA and C^5^-methycytosine) [48,70,71,72,73]. Although these modifications were thought to be irreversible and considered as RNA damages, the identification of several proteins, such as methyltransferase-like gene family (METTL3, METTL13, and METTL14), FTO (fat mass and obesity-associated), and ALKBH5 (AlkB homologue 5) as enzymes that dynamically deposit and oxidatively reverse N^6^-methyladenosine implies a functional role of these modifications [48,67,73,74,75].

6-mA, which is among the most studied modifications, functions in several biological processes such as the regulation of RNA splicing via alteration of the pre-mRNA structure [76], increases transcription termination and control of DNA repair response at double-strand breaks (DSBs) by modulating the formation and accumulation of R-loops (DNA:RNA hybrids), respectively [77,78,79], and increases miRNA biogenesis via the regulation of the DGCR8 protein, part of the miRNA microprocessor complex [73,80].

While some RNA methylations appear to be specific for certain classes, 5-mC occurs on mRNAs, rRNAs, tRNAs, and ncRNAs. 5-mC deposition relies on the RNA-specific subset of S-adenosyl-methionine-dependent methyltransferases, TRDMT1 (also known as DNMT2), and the NSUN1-7 family [73,81,82,83]. Although the function depends on the position and RNA molecule where 5-mC is present, this mark has been generally associated with the regulation of RNA metabolism, in particular with functional and structural RNA stability [73]. For example, on tRNA, it occurs mostly at the junction of the variable loop and the T-stem, where it affects the proper L-shape folding of the tRNA. It also helps to maintain homeostasis and to control translation efficiency and fidelity when occurring in the anticodon loop [72,73,84,85,86]. On rRNA, 5-mC has been described on both rRNA subunits (small and large), and it controls ribosomal synthesis and processing of stress-responsive mRNAs. It affects translation fidelity through altering the structure and conformation of the rRNA [73,87,88]. On mRNA, 5-mC is enriched in 5′- and 3′-UTR regions [73]. It affects RNA maturation, stability, splicing, nuclear export, DNA damage repair, and reprogramming of stem cells, to name a few [72,86,89,90,91,92]. 5-mC has also been detected in a variety of ncRNAs, with a possible function in processing, stability, and interaction of 5-mC readers [73].

Interestingly, the same TET enzymes involved in 5-mC demethylation on DNA are responsible for both mediating the reversal of 5-mC on RNA by functioning as RNA demethylases [89,93] and specifically oxidating 5-mC to 5-hmC [89,94]. The RNA 5-hmC mark has been associated with controlling the transcriptional landscape in embryonic stem cells (ESCs), where it regulates pluripotency and lineage-priming mRNAs for a proper differentiation and cell state transition [73,95,96]. It regulates mRNA half-life and splicing [73]. 5-hmC can be further deaminated by APOBEC to form 5-hmU, which has a role in RNA quality control [64,97,98,99].

#### 3.2.2. RNA Oxidation

Reactive oxygen species (ROS) rapidly oxidize both DNA and RNA, especially at guanine, which has the lowest redox potential among all the bases. Although both nucleic acids are subjected to oxidation, RNA is more susceptible to oxidation due to its single-stranded nature, vicinity to ROS production, and absence of protein interactions [100,101]. Several oxidized forms are generated in RNA, but o^8^G is the most abundant RNA oxidation [101]. Although all its regulatory functions are not completely understood, o^8^G modifies RNA–RNA interactions in a redox-dependent manner by altering the base-pairing properties. This is critical when occurring on miRNAs, where an altered base pairing between the seed region and the mRNA would change the selection of targets [61]. The reader is directed to [64] for a more detailed overview of RNA modifications.

Thus, epigenetic and epitranscriptomic gene regulation shares several enzymes and processes that are able to control all the aspects of the cellular metabolism. It remains to be understood how these enzymes can distinguish between functional modification and DNA/RNA damage. A clear role for glycosylases, especially for the uracil DNA glycosylase family, as key actors for epigenetic control is emerging.

## 4. Uracil DNA Glycosylases and Epigenetic Marks

The uracil DNA glycosylase (UDG) superfamily is deeply involved in the recognition and removal of oxidized and deaminated cytosine products, an essential process for the dynamic control of these bases (Figure 2). TDG recognizes mismatched pyrimidines (G:U and G:T pairs) [102,103]. The first indication of a role for TDG beyond BER was the embryonic lethality caused by TDG deficiency, unlike the knock-outs of the other DNA glycosylases, that mostly have subtle phenotypes [104,105]. Interestingly, it was found that this lethality was associated with epigenetic aberrations and that TDG-dependent DNA repair function is essential for epigenetic maintenance [104,106,107]. TDG has been implicated in transcriptional regulation and chromatin regulation via the interaction with nuclear hormones receptors and transcription factors (retinoic acid and X receptors, c-jun, estrogen receptor) to regulate development, growth, and lineage commitment [108,109,110]. TDG was the first glycosylase to be described as responsible for active demethylation in mammals through the recognition of the TET-oxidized bases, 5-caC and 5-fC [48,56,111]. The discovery that TET-induced oxidation is not limited to 5-mC but it also generates 5-hmU from thymine, together with the identification of specific 5-hmU readers (different protein recruitment for 5-hmU:G and 5-hmU:A pairs), demonstrated a function for 5-hmU besides DNA damage. TDG can recognize 5-hmU:G, but current data suggest that 5-hmU is primarily paired with A, suggesting a role for SMUG1 in epigenetic control [35,47]. SMUG1 is the main 5-hmU DNA glycosylase, removing modified bases in both single-stranded and double-stranded DNA [5,112,113,114]. SMUG1 removes all the oxidative products derived from thymine (5-hmU, 5-carboxyuracil [5-fU], 5-formyluracil [5-caU]) when paired with both A and G [35,115,116,117]. In addition to TDG and SMUG1, a function in epigenetic programming was also shown for MBD4, owing to its ability to recognize both 5-fU and 5-caU [35].

## 5. Epigenetic Marks and DNA Damage/Repair

As mentioned earlier, DNA repair not only influences the deposition/removal of epigenetic marks, but the presence of methylated cytosines can also affect the susceptibility of the specific locus towards DNA damage. This sensitivity depends on the epigenetic marks and on the complexity of the local chemical context. Cancer aetiology involves complex interactions between environmental and hereditary causes. Changes in genomic information are a well-known cause of cancer onset. Like genomic instability, epigenome dysregulation, through DNA methylation, ncRNAs, chromatin remodelling, also plays a key role in tumorigenesis and therapy resistance [118,119,120]. These alterations regulate biological processes such as proliferation, invasion, and senescence. In tumour tissues, cells present distinct patterns of epigenetic modifications, defining heterogeneity at the cellular level [118]. Deregulated DNA/RNA methylation deposition or de novo methylation can result in several pathological conditions such as inflammation and cancer by inactivating gene transcription and altering chromatin stability [118,121,122,123,124]. Abnormal patterns of 5-mC, 5-hmC, and m-6A deposition and eraser dynamics have been linked to cancer onset and chemotherapy resistance. Oncogene activation or silencing of tumour suppressors mediated by 5-mC deposition both on RNA and DNA by alteration in DNMTs expression and/or activity is a common pattern in several tumour settings like breast, bladder, gastric, and lung cancers [73]. TET enzymes appear as both tumour promoters and suppressors [125]. Expression levels of TET proteins, or the modulation of their activity, can derive from 5-mC-mediated silencing of TET loci or changes in the stoichiometry of TET proteins/TET co-factors. TET inactivation and, consequently, reduced demethylation of 5-mC play an important role in determining cancer phenotypes, with a negative correlation regarding the prognosis [126,127]. Loss of TETs is associated with increased levels of G-quadruplexes and R-loops that lead to increased DSBs, especially at immunoglobulin switch regions in B cells, promoting oncogenesis [128]. Conversely, TET activity and the conversion of 5-mC to 5-hmC at DNA damage sites favour DNA repair, preventing chromosome segregation defects [129].

### 5.1. Epigenetic Marks at DNA Damage Sites

Base modifications can affect the rate of DNA decay. For example, 5-mC occurring in specific loci might promote the preferential formation of DNA damage and photoproducts, due to the higher energy absorption of the methylated cytosine [130,131,132,133]. On the other hand, the formation of pyrimidine dimers accelerates the deamination rate of cytosines [134]. 

In general, both 5-mC and 5-fC could affect the rate of spontaneous DNA damage [135,136]. An exception is Alu element methylation, which prevents endogenous DNA damage [137]. Methylation occurring at CpG islands is usually a hot spot for mutations [138,139]. The hydrolytic deamination of cytosine results in uracil, which is easily recognized and repaired by the UDG glycosylases. However, in the presence of 5-mC, the deamination would form thymine, a base naturally occurring on the DNA and more difficult to be detected by TDG, resulting in C-to-T transitions, preventing the maintenance of both methylome and genome integrity [136,140]. The spontaneous hydrolytic deamination rate is also increased at methylated cytosines vs. the unmethylated counterpart [141]. Additionally, 5-mC has an indirect impact on the damage susceptibility through the regulation of higher chromatin structure, nucleosome deposition, occupancy, and stability, as well as transcription factor binding dynamics [136,142,143,144,145]. 5-mC has also been reported to stabilize stalled replication forks, contributing to genome instability [146].

Moreover, 5-hmC may positively affect DNA repair: the accumulation of 5-hmC at DNA damage sites may help the detection of genomic regions that need repair, leading to an inverse correlation between the levels of 5-hmC and the mutation frequency [73,129,138,147]. It facilitates the degradation of stalled replication forks [146]. 5-hmC modification is able, in fact, to control and induce a change in the local chromatin accessibility via decreasing nucleosome affinity by weakening histone 2 and 3 interactions, facilitating the elongation of RNA polymerase II, and increasing unwinding [73,148]. All these changes could serve as a recruitment platform for DNA damage repair enzymes at the repair spot, as exemplified by the role of 5-hmC as a recruitment marker at stalled replication forks for the DNA glycosylases TDG and SMUG1, which creates an AP site for the subsequent action of APE1 [146]. However, these alterations in DNA helix stability and nucleosome density may also induce the formation of R-loops. 5-hmC directly promotes R-loop and G-quadruplex formation in active genes both in vitro and in vivo. Endogenous R-loop levels are reduced after the depletion of TET enzymes, suggesting that their activity favours R-loop formation in the absence of transcriptional changes [125,128,149]. Both 5-hmC and R-loops have a very similar distribution at intragenic regions, reaching a peak at the transcription termination site (TTS), suggesting an intriguing hypothesis for an involvement of TET enzymes in transcription termination via R-loop formation. In contrast, an inverse correlation between 5-hmC and R-loops is present at the transcription start site (TSS) regions, where R-loops are abundant probably by other chromatin and DNA features. This suggest that 5-hmC modification is not necessary for co-transcriptional DNA:RNA hybridization and R-loop formation at TSS regions, but R-loops formed in 5-hmC regions impact gene expression in stem cells [125]. The disruption of R-loop homeostasis causes genomic instability [150] via the generation of barriers to fork progression due to conflicts between transcription and replication machineries that can lead to DNA damage in form of DSBs [151,152]. A small but significant correlation was found between DSBs, R-loops, and 5-hmC, suggesting that the 5-hmC region can be prone to DNA damage as well due to R-loop formation [125]. Although these unscheduled R-loops may threaten DNA integrity in 5-hmC-rich regions, a possible regulatory role could be envisioned for the regulated formation of R-loops in the same regions, such as the TET-regulated transcription termination [125]. 

### 5.2. RNA Epigenetic Marks and DNA Damage Response

Modifications occurring on RNA, such as methylation at 6-A and 5-C and its hydroxymethylation, can also affect DNA damage response and repair [78,79,149]. This is not surprising considering that several RNA species (i.e., DNA damage response and damage-induced RNAs, DNA:RNA hybrids) have recently been associated with DNA damage response and described as essential for achieving the correct DNA repair [153]. For example, the recruitment of DNMT2 at damage sites introduces 5-mC on DNA:RNA hybrids, a signal to promote early DNA repair via the recruitment of DNA repair enzymes. The R-loops need to be resolved for the correct completion of the DNA repair, and since 5-mC increases the RNA’s stability, 5-mC has to be removed for a timely resolution of the R-loops in the late phase of DNA repair. FMRP (Fragile X messenger ribonucleoprotein 1), a newly discovered 5-mC reader, is recruited at the damaged sites via direct interaction with DNMT2 and contributes to the demethylation of 5-mC to 5-hmC at the R-loops via TET1 oxidation, preventing R-loop accumulation and stabilization at the damaged sites, ultimately favouring DSBs repair [91,149]. Apart from its role in 5-mC demethylation, FMRP can also bind 6-mA modified RNA, modulating its stability and nuclear export, suggesting a tight control for RNA writers and readers depending on the cellular context [149]. Another modification occurring on R-loops upon DNA damage and at DSBs sites is 6-mA, which serves as a recruitment platform for downstream repair factors such as RAD51 and BRCA1 and coordinates the resolution of the R-loops [78,153]. Interestingly, ADP-ribosylation, a well-known modification for DNA and proteins involved in a variety of biological processes such as DSB repair, was recently described as a dynamic modification in RNA [154]. Further studies are needed for describing its functions in vivo, but it is tempting to speculate its possible role in DNA damage response [154].

## 6. SMUG1 Canonical (DNA Repair) and Non-Canonical (RNA Quality Control for Non-Coding RNAs) Functions

Eight DNA glycosylases, with overlapping substrate specificities, handle various pyrimidine-derived lesions in mammalian cells. Four of them recognize uracil: UNG, SMUG1, TDG, MBD4 (in addition to mismatch repair) [155]. Simpler organisms like *Caenorhabditis elegans* only need one; non-vertebrates appear to have either UNG- or SMUG1-type enzymes and have a high degree of functional redundancy [156]. In ‘higher’ species, both enzymes are present with specialised functions through the cell cycle [157], distinct sub-cellular localization, and protein–protein interactomes. Thus, the disruption of just one is often sufficient to confer a clear phenotype in cells, as exemplified by recent papers validating that the inhibition or downregulation of one DNA glycosylase is sufficient to inhibit cancer cell proliferation [26,112,158,159,160]. In the case of telomere stability, it seems that perturbation of just one DNA glycosylase is sufficient to induce telomere shortening and dysfunction, alone or in combination with DNA damaging agents [161,162,163,164,165]. Thus, uracil DNA glycosylases are not functionally redundant, cannot fully substitute for each other in case of pharmaceutical inhibition, and have specialized cellular roles.

The complex interconnection between epigenetic modification, DNA damage response, and genome integrity can be exemplified by SMUG1. Originally identified as a DNA glycosylase recognizing and excising 5-hydroxymethyl-2’-deoxyuridine residues in DNA [166], SMUG1 also recognizes a number of pyrimidine-derived lesions (uracil, 5-hydroxymethyluracil, 5-formyluracil, and 5-fluorouracil in single- and double-stranded DNA). However, several observations suggest that SMUG1 might have additional functions that are well outside of the BER pathway (Figure 3).

First, in mammalian cells, its activity on substrates packaged in nucleosomes is reduced by almost an order of magnitude compared to naked DNA [113,167,168]. Second, SMUG1 localizes to nucleoli [169] and Cajal bodies [97], nuclear structures that are key to the processing and assembly of various ribonucleoprotein complexes such as ribosome subunits and telomerase holoenzyme. In these locations, SMUG1 was found to interact directly with proteins involved in RNA maturation, i.e., the pseudouridine synthase dyskerin [97], which is also a core component of the telomerase complex. Moreover, SMUG1 was shown to recognize and excise damaged bases from RNA, i.e., 5-hydroxymethyluridine (Figure 4) [97,170]. *SMUG1* knockdown leads to the accumulation of 5-hydroxymethyluridine in RNA, accompanied by a depletion of ribosomal RNA subunits [97], strongly suggesting that SMUG1 might have a biologically relevant specialized function in RNA quality control. SMUG1 is also required for the maturation of *hsa-let-7b-5p* [171] and the maintenance of the telomerase RNA component (*hTERC*) in mammalian cells [170], apparently by regulating levels of modified RNA bases at the 3′-end of *hTERC* RNA, a region known to be methylated at C323 and C445 [172]. In cells, SMUG1 deficiency led to a reduction in telomerase activity (although TERT expression was not compromised), telomere attrition, perturbed localization of Dyskerin, and fragile telomeres [170]. These data suggest that SMUG1 might regulate telomere stability, which may in turn limit cancer cell growth.

Concomitant with a role for SMUG1 in RNA quality control, its role on DNA was recently expanded as one of the DNA glycosylases involved in the active demethylation processes after the recognition of 5-hmU and replication fork stability (Figure 4) [30,57,146]. Its capability to recognize 5-hmU and its recruitment at the actively transcribing DNA polymerase II complex at the *hTERC* promoter suggest a possible involvement of SMUG1 in epigenetic regulation [170].

## 7. Conclusions and Future Perspectives

Scientists are just now starting to unveil the complexity with which modified nucleic acid bases, for example, 5-hmU, can impact many cellular processes and have been suggested as cancer biomarkers (Figure 5).

Although the recognition of hmU is likely mainly carried out by the DNA glycosylases, the timing and specific outcomes are determined by additional layers of regulation. Matchmaking proteins may be required to guide the enzymes to various substrates (DNA, lncRNA, miRNA) or genomic locations. A deeper comprehension of how these processes are regulated and connected in time and space for development is essential. This will require the development of new technologies that are able to trace highly dynamic, rare base modifications at single-nucleotide resolution through time and space.

## Figures and Tables

**Figure 1 ijms-24-10307-f001:**
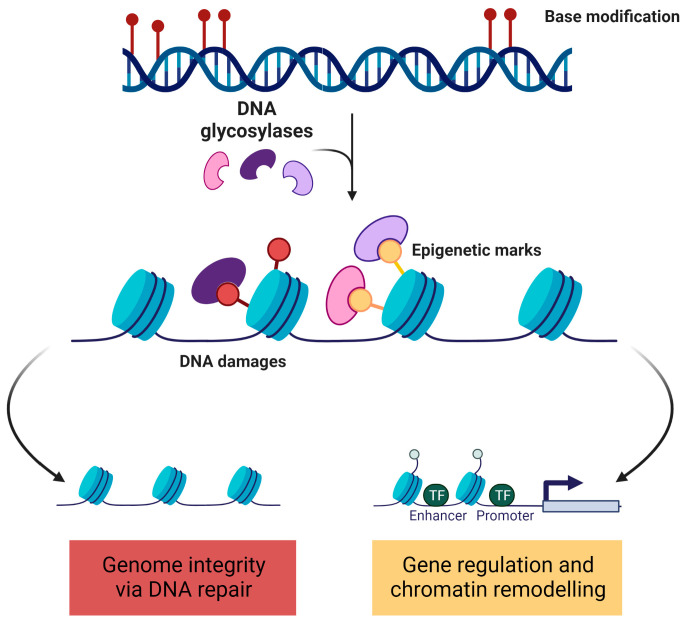
**DNA glycosylases act as DNA sensors for discriminating between DNA damages and epigenetic marks.** The same chemical modification can act as DNA damage or epigenetic modification. DNA glycosylases need to discriminate the nature of these modifications in order to restore genome integrity via DNA repair or to regulate gene expression and chromatin remodelling. The figure has been generated using Biorender.com.

**Figure 2 ijms-24-10307-f002:**
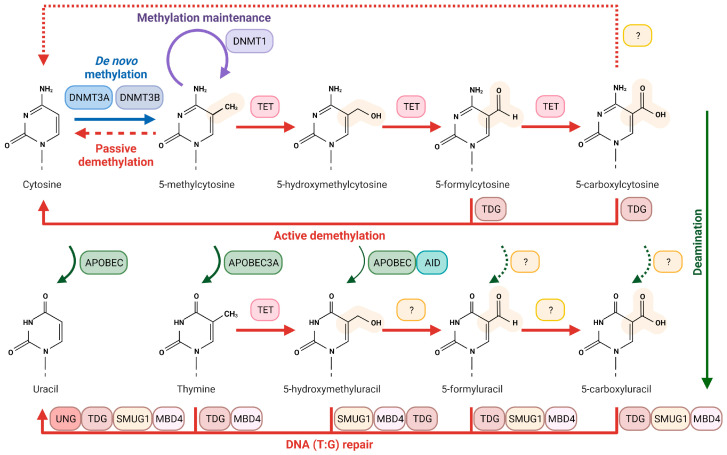
**Oxidized cytosines, demethylation and deamination pathways.** DNMT3A and DNMT3B catalyse the methylation at the 5- position of cytosine, forming 5-mC. This modified base is then subsequently oxidized by TET enzymes to 5-hmC, 5-fC, and 5-caC. 5-fC and 5-caC can be actively recognized by TDG and replaced to cytosine via BER. APOBEC enzymes are responsible for the deamination of these bases. 5-hmU can be generated by deamination from 5-hmC (in a lower rate) and by direct TET-oxidation of thymine. 5-hmU mismatches are recognized mainly by SMUG1, but they can be detected by TDG and MBD4 glycosylases as well. Other direct mechanisms for active demethylation are not well-established. A direct decarboxylation has been demonstrated, although the enzyme responsible for this activity have not been characterize yet (indicated with a ‘?’ in the figure). A passive demethylation mechanism involves the loss of methylation during the replication process. Still controversial is the deamination from 5-fC and 5-caC to 5-fU and 5-caU, respectively. The further oxidation from 5-hmU to 5-fU and 5-caU and the enzymes involved in this process have not been characterized. 5-mC, 5-methylcytosine; 5-hmC, 5-hydroxymethylcytosine; 5-fC, 5-formylcytosine; 5-caC, 5-carboxylcytosine; 5-hmU, 5-hydroxymethyluracil; 5-fU, 5-formyluracil; 5-caU, 5-carboxyluracil. The figure has been generated using Biorender.com.

**Figure 3 ijms-24-10307-f003:**
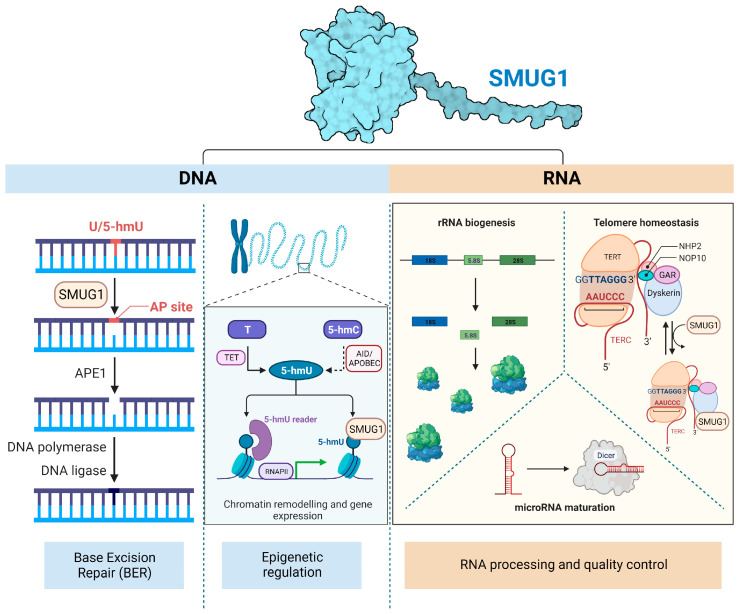
**SMUG1 functions on DNA and RNA.** On DNA, the classical role of SMUG1 as one of the uracil glycosylases for the BER has been recently expanded though the possible role in epigenetic regulation via recognition of 5-hmU. SMUG1 functions in RNA processing and quality control with functional relevance for ribosomal RNA biogenesis, telomere maintenance, and microRNA maturation. SMUG1 structure was predicted with AlphaFold (Uniprot ID: A0A024RAZ8). Telomerase reverse transcriptase, TERT; telomerase RNA component, *TERC*. The figure has been generated using Biorender.com.

**Figure 4 ijms-24-10307-f004:**
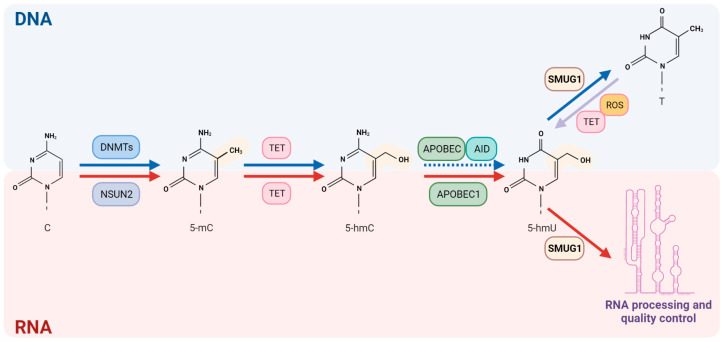
**SMUG1 role in recognition of 5-hmU on both DNA and RNA.** SMUG1 recognizes 5-hmU on both nucleic acids. It restores the unmodified base on DNA via BER that can be derived from deamination of 5-hmC and oxidation of thymine. The recognition of 5-hmU on RNA by SMUG1 starts RNA processing and quality control processes for several non-coding RNAs. The figure has been generated using Biorender.com.

**Figure 5 ijms-24-10307-f005:**
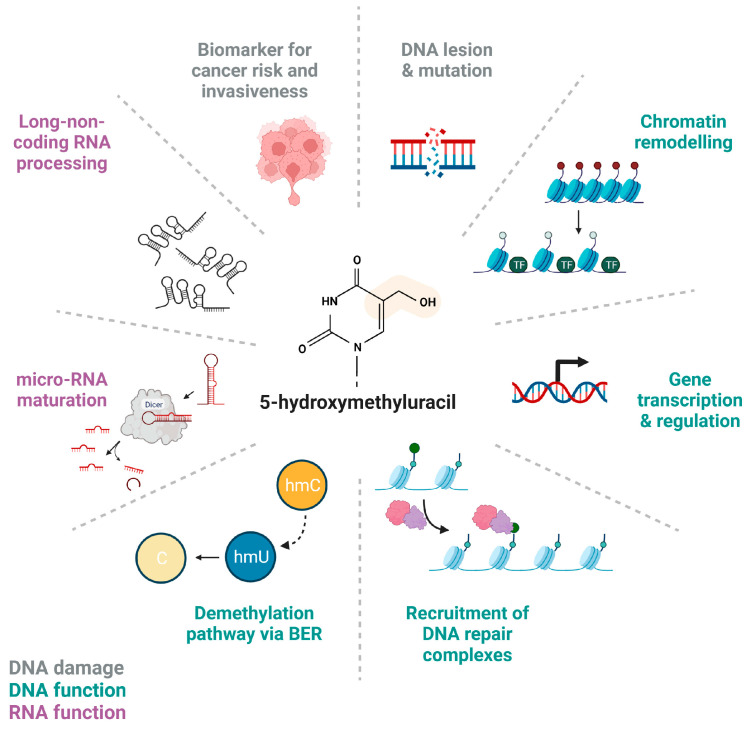
**5-hydroxymethyluracil in DNA and RNA.** Schematic representation of known and proposed functions of 5-hmU in nucleic acids and as DNA lesion. 5-hmU can lead to mutation (DNA damage), if unrepaired, and its level in blood can be used as biomarker for cancer risk and invasiveness. On DNA, it affects gene transcription and regulation, chromatin remodelling, and recruitment of DNA repair complexes. On RNA, it impacts the processing and the maturation of long-non-coding and micro-RNAs. The figure has been generated using Biorender.com.

## Data Availability

Not applicable.

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
