# Peer review of "DNA Glycosylases Define the Outcome of Endogenous Base Modifications"

_ijms, 2023, doi:10.3390/ijms241210307_

Round 1
Reviewer 1 Report
Dear colleagues,
Thank you for giving me the opportunity to read and provide a review on the article of Lirussi and Nielsen about DNA glycosylases.
I have not much to comment about the present review. It is well detailed and the authors have a good knowledge of what they are talking about. I have however one or two general questions/comments but I understand that the answer may not be found in the literature: What is known about the interaction with nucleosome occupancy or depleted regions? The second relates to how conserved are those mechanisms throughout evolution (L303: the description was made in mammals, but this gene, and the other, are found in far many species). Even though one can understand that the purpose of this review is very mechanistic, considering how these mechanisms were set up, or could have taken place during the course of evolution could however bring this already good review to another level. For example, I searched the bibliography and see that some of the genes considered here also exist in Bacteria (ref 154). Where do these genes come from and how did they specialized would be an interesting question, but (not extensively) screening the literature seems to provide too few hints to develop this topic in a review. It could be interesting questions to consider.
A few suggestions:
L47 and around: talking about Uracil, but need to read L76 to see why U is present in DNA (for less specialized readers).
L96: A very short paragraph.
L117: APOBEC why not using the alphanumerical order?
L406: FMRP not yet introduced.
A few typographic mistakes noted (not an exhaustive list):
L19: the ??? of
L47: 5-hyDroxy
L330: patterNs
L369: by weakenING
L403: The R-loopS needS
L430: is sufficient TO inhibit
L453: Secondly --> Second (Like "First, l451)
Fig5 caption: 2 final dots.
L502: Funding: Funding:
Author Response
Reviewer 1
Thank you for giving me the opportunity to read and provide a review on the article of Lirussi and Nielsen about DNA glycosylases.
I have not much to comment about the present review. It is well detailed and the authors have a good knowledge of what they are talking about.
We would like to thank the reviewer for his/her comments.
- I have however one or two general questions/comments but I understand that the answer may not be found in the literature: What is known about the interaction with nucleosome occupancy or depleted regions?
We agree with the reviewer that this is one of the questions that the field needs to tackle. As it is known, the epigenetic information including DNA methylation, nucleosome positioning, histone modification and 3D chromatin conformation has important associations with gene function and expression patterns. Nucleosome mobility, disassembly, and destabilization have a profound impact on gene expression regulation and other DNA-dependent processes. The nucleosome position is determined by multi-factors as DNA sequence and epigenetic modifications, transcription factors, RNA polymerase II, and histone modifications and variants, to mention a few. The available literature suggests that the nucleosome occupancy at specific loci may affect DNA glycosylase binding and activity on these DNA modifications, due to accessibility to the DNA itself and possibly masking the DNA mark. However, the position of these modifications at genome-wide level is essential to prove this hypothesis. The complex and combined effect of DNA modifications such as 5-hmU, 5-fU and 5-caU and nucleosome position/occupancy at determined genomic location hasn’t been properly investigated due to the lack of single-nucleotide mapping methods for these DNA modifications. Combining the information of the epigenetic landscape at a genome-wide level through new NGS technology with nucleosome-omics will provide a new perspective on the epigenetic code, 3D genome landscape and gene regulation.
- The second relates to how conserved are those mechanisms throughout evolution (L303: the description was made in mammals, but this gene, and the other, are found in far many species). Even though one can understand that the purpose of this review is very mechanistic, considering how these mechanisms were set up, or could have taken place during the course of evolution could however bring this already good review to another level. For example, I searched the bibliography and see that some of the genes considered here also exist in Bacteria (ref 154). Where do these genes come from and how did they specialized would be an interesting question, but (not extensively) screening the literature seems to provide too few hints to develop this topic in a review. It could be interesting questions to consider.
We agree with the reviewer that understanding how these enzymes acquired specialized functions during evolution would be very interesting, especially since these genes are present in different domains, from bacteria and archaea to eukarya. Interestingly, the number of DNA glycosylases changes dramatically when considering bacteria, archaea and higher species, having “lower” species with few glycosylases that are able to detect a wider spectrum of DNA damages. Unfortunately, to our knowledge, not many specific studies about epigenetic marks and DNA glycosylases in bacteria and archaea have been done so far, leaving the question open about how these specialized functions occurred during evolution.
- A few suggestions:
L47 and around: talking about Uracil, but need to read L76 to see why U is present in DNA (for less specialized readers).
As suggested by this Reviewer, we have now introduced uracil in lines 47-49. The new sentence can be now read as: “Some examples of these are oxidation of guanine, cytosine deamination to uracil, and the oxidation intermediates of 5-hydroxymethylcytosine (5-hmC) and 5-hydroxymethyluracil (5-hmU).”
L96: A very short paragraph.
We have now combined this paragraph with the next one, describing the cancer syndromes associated with mutations in BER genes.
L117: APOBEC why not using the alphanumerical order?
We now changed to APOBEC3A/APOBEC3B (line 129).
L406: FMRP not yet introduced.
As suggested by this Reviewer, we have now introduced FMRP in lines 417-421. The new sentence can be now read as: “FMRP (Fragile X messenger ribonucleoprotein 1), a newly discovered 5-mC reader, is recruited at the damaged sites via direct interaction with DNMT2 and contributes to the demethylation of 5-mC to 5-hmC at the R-loops via TET1 oxidation, preventing R-loop accumulation and stabilization at the damaged sites, and ultimately favoring DSBs repair [89, 147].”
- A few typographic mistakes noted (not an exhaustive list):
L19: the ??? of
L47: 5-hyDroxy
L330: patterNs
L369: by weakenING
L403: The R-loopS needS
L430: is sufficient TO inhibit
L453: Secondly --> Second (Like "First, l451)
Fig5 caption: 2 final dots.
L502: Funding: Funding:
We apologize for the typographic mistakes. We now read carefully through the paper again correcting the typos indicated in the list and the other ones we could detect.

Reviewer 2 Report
Dear Editors and Authors,
The manuscript “DNA glycosylases define the outcome of endogenous base modifications” by Lirussi L. and Nilsen L. is an interesting and valuable review about the role of DNA glycosylases’ specificity and selectivity in the (epi)genetic regulation. The review is well written, but I have some minor comments and recommendations.
It would be nice to expand the abstract as much as possible. For example, the conclusion and most of the text is devoted to the role of 5-hmU in DNA and RNA, but it was not mentioned in the abstract.
“SBS, single break substitution; DSB, double strand break” – should be added to Abbreviations
96-98: “As many base modifications change base pairing properties, and therefore have mutagenic capacity, [9], it was somewhat surprising that knockout mice deficient in DNA-glycosylases, generally, are not prone to develop spontaneous cancer [10-12].” – Here it would be more interesting for readers to write in more detail about works [9-12]. Please expand.
107-108: “Whole genome sequencing of cancer genomes has revealed mutation signatures associated with defects in BER [19].” – What exactly is this work about? Including what types of cancers were considered in this work? Please expand.
116-123: What types of tumors were considered in the works [22,23]?
167, 174, 222, 224: What do the signs (i) and (ii) indicate?
201: “Although methylation does not alter the Watson-Crick base pairing” – reference is needed.
414-416: “Interestingly, ADP-ribosylation, a well-known modification for DNA and proteins involved in a variety of biological processes as DSB repair, was recently described as dynamic modification in RNA.” – reference is needed.
Author Response
Reviewer 2
The manuscript “DNA glycosylases define the outcome of endogenous base modifications” by Lirussi L. and Nilsen L. is an interesting and valuable review about the role of DNA glycosylases’ specificity and selectivity in the (epi)genetic regulation. The review is well written, but I have some minor comments and recommendations.
We would like to thank the reviewer for his/her comments.
- It would be nice to expand the abstract as much as possible. For example, the conclusion and most of the text is devoted to the role of 5-hmU in DNA and RNA, but it was not mentioned in the abstract.
We agree with the Reviewer and we have now added an extra sentence in the abstract about the role of 5-hmU in nucleic acids. Unfortunately, we had very few words allowed due to word limitations for the abstract. The new sentence (lines 21-24) is: “We will also describe how epigenetic marks, with a special focus on 5-hydroxymethyluracil, can affect the damage susceptibility of nucleic acids and conversely how DNA damage can induce changes in the epigenetic landscape by altering the pattern of DNA methylation and chromatin structure.”
- “SBS, single break substitution; DSB, double strand break” – should be added to Abbreviations
We thank the Reviewer for noticing it. We have now added SBS and DSBs in the abbreviation list (lines 34-35: “SSB, Single-base substitution; DSBs, DNA double-strand breaks.”)
- 96-98: “As many base modifications change base pairing properties, and therefore have mutagenic capacity, [9], it was somewhat surprising that knockout mice deficient in DNA-glycosylases, generally, are not prone to develop spontaneous cancer [10-12].” – Here it would be more interesting for readers to write in more detail about works [9-12]. Please expand.
As suggested by this Reviewer, we added a sentence about DNA glycosylases-deficient mice and formation of cancer. The new paragraph (lines 101-109) can be now read as: “As many base modifications change base pairing properties, and therefore have mutagenic capacity [9], it was somewhat surprising that knockout mice deficient in DNA-glycosylases, generally, are not prone to develop spontaneous cancer [10-12]. Unexpectedly, these knock-out mice present mild to moderately elevated mutation frequencies under normal physiological conditions. In some models, like the UNG knock-out mice, there was some spontaneous tumour formation but at low penetrance and late onset [11-12]. The absence of a tumour prone phenotype supports the hypothesis that DNA glycosylases have evolved specialized functions beyond DNA repair associated with gene regulation, replication, and chromatin remodelling [10-12].
- 107-108: “Whole genome sequencing of cancer genomes has revealed mutation signatures associated with defects in BER [19].” – What exactly is this work about? Including what types of cancers were considered in this work? Please expand.
The work from Karolak et al., [19] describes a new method for inferring mutational signatures based on DNA structural parameters. The authors demonstrate that DNA conformation (base pair, base pair step, and minor groove width) can accurately predict mutations occurring due to DNA repair failures, APOBEC cytosine deaminase and exposure to mutagens. They evaluated single-nucleotide variants (SNV) from 30 cancer types and the data were collected and extracted from The Cancer Genome Atlas (TCGA), the International Cancer Genome Consortium (ICGC), and individual whole-genome sequencing (WGS) of medulloblastoma subtypes. Indels or structural variations were not considered in this study. We modified the sentence (lines 115-120), adding an additional reference for BER mutational signatures. The new sentence can be read as: “Whole genome sequencing of cancer genomes has revealed mutation signatures associated with defects in BER, due to compromised DNA glycosylase activity and by shaping mutation signatures connected with AID/APOBEC family enzymes [19, 20].”
- 116-123: What types of tumors were considered in the works [22,23]?
In Burns et al. 2013 Nature Genetics [22], the authors analyzed 19 different cancer types and showed that APOBEC3B is upregulated and its preferred sequence is mutated in six distinct cancers (bladder, cervix, lung [adenocarcinoma and squamous cell carcinoma, head and neck, and breast) [22]. In Burns et al., 2013 Nature [23], the authors focused on breast cancer cell lines and tumors [23].
6.167, 174, 222, 224: What do the signs (i) and (ii) indicate?
These signs indicate a list of two elements where (i) stands for first and (ii) for second. To avoid any confusion, we deleted them from the text.
- 201: “Although methylation does not alter the Watson-Crick base pairing” – reference is needed.
As suggested by the Reviewer, we added a new reference (ref 60 – line 213) in the sentence: “Although methylation does not alter the Watson-Crick base pairing [60], cytosine methylation affects DNA secondary structure in C-rich sequences through changes in DNA hydrophobicity, steric hindrance and in the mechanical DNA properties [29].”
- 414-416: “Interestingly, ADP-ribosylation, a well-known modification for DNA and proteins involved in a variety of biological processes as DSB repair, was recently described as dynamic modification in RNA.” – reference is needed.
We are sorry for the confusion. The reference was indicated in the next sentence. Now we added it also at the end of this sentence as well (line 428): “Interestingly, ADP-ribosylation, a well-known modification for DNA and proteins involved in a variety of biological processes as DSB repair, was recently described as dynamic modification in RNA [154]. Further studies are needed for describing its functions in vivo, but it is tempting to speculate its possible role in DNA damage response [154].”

Reviewer 3 Report
The review by Lirussi L. et al. entitled "DNA glycosylases define the outcomes of endogenous base modifications" reports endogenous DNA damage and epigenetic modifications, as well as their functions. In despite of identical chemical modifications, endogenous DNA damage and epigenetic modifications exerting different functions. The authors also accurately distinguish between epigenetic marks and DNA damage to ensure proper repair and maintenance of epigenomic integrity. The specificity and selectivity of the recognition of these modified bases relies on DNA glycosylases.
The review is comprehensive, clear, well structured and agreeable to read, I suggest to consider it for publication after the addition of some legends.

Author Response
Reviewer 3
The review by Lirussi L. et al. entitled "DNA glycosylases define the outcomes of endogenous base modifications" reports endogenous DNA damage and epigenetic modifications, as well as their functions. In despite of identical chemical modifications, endogenous DNA damage and epigenetic modifications exerting different functions. The authors also accurately distinguish between epigenetic marks and DNA damage to ensure proper repair and maintenance of epigenomic integrity. The specificity and selectivity of the recognition of these modified bases relies on DNA glycosylases. The review is comprehensive, clear, well structured and agreeable to read, I suggest to consider it for publication after the addition of some legends.
We would like to thank the reviewer for his/her comments.
As suggested by this Reviewer, we have now implemented the legends for Figure 1 and Figure 5. The new legends can now be read as:
Figure 1: DNA glycosylases act as DNA sensors for discriminating between DNA damages and epigenetic marks. The same chemical modification can act as DNA damage or epigenetic modification. DNA glycosylases need to discriminate the nature of these modifications in order to restore genome integrity via DNA repair or to regulate gene expression and chromatin remodelling. The figure has been generated using Biorender.com.
Figure 5: 5-hydroxymethyluracil in DNA and RNA. Schematic representation of known and proposed functions of 5-hmU in nucleic acids, and as DNA lesion. 5-hmU can lead to mutation (DNA damage), if unrepaired, and its level in blood can be used as biomarker for cancer risk and invasiveness. On DNA, it affects gene transcription and regulation, chromatin remodelling and recruitment of DNA repair complexes. On RNA, it impacts the processing and the maturation of long-non-coding and micro- RNAs. The figure has been generated using Biorender.com.
